# Amputee Fall Risk Classification Using Machine Learning and Smartphone Sensor Data from 2-Minute and 6-Minute Walk Tests

**DOI:** 10.3390/s22051749

**Published:** 2022-02-23

**Authors:** Pascale Juneau, Natalie Baddour, Helena Burger, Andrej Bavec, Edward D. Lemaire

**Affiliations:** 1Ottawa Hospital Research Institute, Ottawa, ON K1Y 4E9, Canada; elemaire@ohri.ca; 2Department of Mechanical Engineering, University of Ottawa, Ottawa, ON K1N 6N5, Canada; nbaddour@uottawa.ca; 3University Rehabilitation Institute, University of Ljubljana, 1000 Ljubljana, Slovenia; helena.burger@ir-rs.si (H.B.); andrej.bavec@ir-rs.si (A.B.); 4Faculty of Medicine, University of Ljubljana, 1000 Ljubljana, Slovenia; 5Faculty of Medicine, University of Ottawa, Ottawa, ON K1N 6N5, Canada

**Keywords:** 6MWT, 2MWT, foot strike detection, amputee, LSTM, random forest, fall risk classification, artificial intelligence, smartphone

## Abstract

The 6-min walk test (6MWT) is commonly used to assess a person’s physical mobility and aerobic capacity. However, richer knowledge can be extracted from movement assessments using artificial intelligence (AI) models, such as fall risk status. The 2-min walk test (2MWT) is an alternate assessment for people with reduced mobility who cannot complete the full 6MWT, including some people with lower limb amputations; therefore, this research investigated automated foot strike (FS) detection and fall risk classification using data from a 2MWT. A long short-term memory (LSTM) model was used for automated foot strike detection using retrospective data (n = 80) collected with the Ottawa Hospital Rehabilitation Centre (TOHRC) Walk Test app during a 6-min walk test (6MWT). To identify FS, an LSTM was trained on the entire six minutes of data, then re-trained on the first two minutes of data. The validation set for both models was ground truth FS labels from the first two minutes of data. FS identification with the 6-min model had 99.2% accuracy, 91.7% sensitivity, 99.4% specificity, and 82.7% precision. The 2-min model achieved 98.0% accuracy, 65.0% sensitivity, 99.1% specificity, and 68.6% precision. To classify fall risk, a random forest model was trained on step-based features calculated using manually labeled FS and automated FS identified from the first two minutes of data. Automated FS from the first two minutes of data correctly classified fall risk for 61 of 80 (76.3%) participants; however, <50% of participants who fell within the past six months were correctly classified. This research evaluated a novel method for automated foot strike identification in lower limb amputee populations that can be applied to both 6MWT and 2MWT data to calculate stride parameters. Features calculated using automated FS from two minutes of data could not sufficiently classify fall risk in lower limb amputees.

## 1. Introduction

The six-minute walk test (6MWT) is a sub-maximal movement assessment used to evaluate aerobic capacity and mobility [1]. The 6MWT was originally developed for those with chronic respiratory or cardiovascular disease [2] but is now used to assess a number of populations, commonly older adults, people who have suffered a stroke, people with Parkinson’s disease, and lower limb amputees [3,4,5,6]. The test is brief and requires minimal set up and space (minimum 12 m) [7]. Participants are allowed to walk with mobility aids, stopping is allowed if needed, and distance walked moderately correlates with more complex aerobic capacity tests, such as VO_2_ max, minimizing the burden for patient and clinician [8,9,10,11,12]. However, some people are unable or unwilling to complete a 6MWT due to limited mobility. The two-minute walk test (2MWT) is a similar assessment to the 6MWT but only requires two minutes of walking. Distances walked during a 2MWT correlate well with distance walked during a 6MWT, so the 2MWT is a viable alternative [13,14,15].

Recent research has sought to extract richer information from 2MWT and 6MWT using artificial intelligence (AI). The Ottawa Hospital Rehabilitation Centre (TOHRC) Walk Test app collects acceleration, angular velocity, and orientation data from a smartphone during walking. These smartphone signals can then be used to automatically detect foot strikes (FS). FS detection is a necessary step in gait analysis because a FS defines the start and end of a gait cycle and can be used to calculate stride parameters such as step time and cadence. Capela et al. [16] demonstrated that smartphone signals collected using the TOHRC Walk Test app could be used as input data in a rule-based algorithm to identify FS and calculate stride parameters for a 2MWT or 6MWT in able-bodied participants. Capela et al. [17] further examined that, when applied to 6MWT smartphone data of healthy older adults, the rule-based algorithm identified FS with 99.9% accuracy. With the TOHRC Walk Test App, stride parameters are calculated immediately after completion of the walk test to provide real-time reporting to a clinician. However, when a similar rule-based algorithm was applied to lower limb amputee gait data, accuracy decreased to 87.0% and offline error correction was required [18]. Amputee gait differs markedly from healthy adults, which can make it difficult for AI algorithms to automatically detect steps.

Juneau et al. [19] developed a novel long-short term memory (LSTM) deep learning approach for automated FS detection in lower limb amputees. The LSTM was trained on filtered smartphone signals collected from the TOHRC Walk Test app during a 6MWT. FS and non-FS events were classified with 99.0% accuracy, using offline error correction. Stride parameters calculated from the automated FS were equivalent to manually labeled results for most participants. Stride parameters are not an outcome measure that is typically available from a 6MWT, demonstrating that clinical outcome measures can be calculated using automated FS from 6MWT data.

Lower limb amputees typically have greater instability and higher variability during walking than healthy older adults, leading to elevated risk of falling [20]. Due to this, people may prefer to complete a 2MWT instead of a 6MWT. However, the deep learning automated FS approach has not yet been validated with 2MWT data. The LSTM FS identification model described in [19] was originally trained on data from a 6MWT; however, it is possible that a new algorithm specific to 2MWT data is required. Therefore, this study evaluated LTSM FS identification accuracy on 2MWT data after the model was trained on signals from a 6MWT or signals from a 2MWT, stride parameters were compared between automated FS and manually labeled FS, and fall risk classification was assessed using step-based features from automated and manually labeled FS. A successful 2MWT model would enhance the range of applications for this smartphone-based assessment approach, thereby enhancing access and immediacy of movement-based analyses for people with physical disabilities.

## 2. Materials and Methods

### 2.1. Recruitment and Participants

A convenience sample of 93 transtibial, transfemoral, and bilateral lower limb amputees were recruited from the University Rehabilitation Institute (Ljubljana, Slovenia) and gave informed consent for this study (Table 1). This research was approved by the Ethic Committee of the University Rehabilitation Institute, Slovenia (# 46/2018) and re-approved for an additional 30 participants (# 27/2019).

Each participant’s self-reported fall history was used to classify participants as no fall risk or fall risk. Participants were considered a fall risk if they reported falling at least once in the past six months prior to testing. The inclusion criteria were: transtibial or higher amputation; ability to walk with single cane, two crutches, or without any walking aids; minimum of six months post-amputation; had a functional prosthesis; no wounds on the residual limb; and was willing to participate. Excluded trials were due to unknown fall risk status (8 participants) and cell phone affixed to the side of the hip instead of lower back (5 participants).

### 2.2. Data Collection

Each participant completed a 6MWT along a 20 m hallway with an Android smartphone affixed to the posterior pelvis (Figure 1). The 6MWT was video recorded using a second Android smartphone for each participant. The TOHRC Walk Test app collected smartphone acceleration (m/s), angular velocity (rads/s), and smartphone orientation at an average of 50 Hz [17].

### 2.3. Pre-Processing

#### 2.3.1. Filtering and Signal Processing

Raw accelerometer data, gyroscope data, smartphone orientation, and timestamps were exported for pre-processing and were imported into MATLAB 2020b. Since smartphone signals have a variable sampling rate, each signal was re-interpolated at 50 Hz for a total of 18,049 data points per signal per participant over the 6MWT. Signals were then filtered with a fourth-order zero-lag Butterworth low pass filter with a 4 Hz cut-off frequency. 

To determine if the automated FS detection model from [19] can predict FS in 2MWT data, a simulated 2MWT dataset was created. The first two minutes of the 6MWT trial was determined to replicate most closely that of a 2MWT, so the first two minutes of each participant’s trial was exported for a total of 6000 data points per participant.

#### 2.3.2. Manual Ground Truth Labeling

Two assistants manually identified and labeled ground truth steps prior to model training. The two class labels were label 0 (no foot strike) and label 1 (foot strike) and were identified using the following procedure. Linear acceleration signals over time were graphed and anterior–posterior (AP) acceleration signal peaks were identified (Figure 2). Usually, FS events correspond with AP acceleration peaks that are followed by a peak in vertical acceleration. Acceleration signals matching this pattern were visually identified and a FS event was recorded at the timestamp of AP signal peaks immediately followed by a vertical signal peak. Timestamps were confirmed using participant video. An agreement of the two assistants was required in cases where the AP peak was not well defined or in cases of multiple peaks to select the most appropriate location for the FS event. All other timestamps were therefore labeled as “no foot strike”.

### 2.4. Foot Strike Classification Models

An LSTM deep learning approach was developed and evaluated for automated FS detection in [19]. The model was written and evaluated in Python 4.2. The LSTM layer was imported from Keras [21] and several hyperparameter combinations were evaluated. The LSTM from [19] with the best performance was subsequently used for this analysis. The LSTM had 100 hidden nodes in the LSTM layer, the dense layer had 50 hidden nodes, a batch size of 64 and a dropout value of 0.4 were used. Smartphone orientation, XYZ coordinates for raw and linear acceleration (m/s^2^), and angular velocity (rads/s) from the full 6MWT and the first two minutes of walk test data were the input data.

Smartphone signals were formatted into data windows prior to model input. Each window spanned 15 frames (0.3 s) before the class label to 15 frames after the label. For the first 15 data points, 30 frames after the class label were used. Similarly, the previous 30 frames were used for the final 15 data points. The 31-frame window size (i.e., 15 before, labeled frame, 15 after) was selected to minimize the likelihood of multiple FS events occurring within the same window of signal data.

The LSTM FS identification model was trained on the full six minutes of data (6M-FS model) and then re-trained on the first two minutes from the 6MWT, to approximate a 2MWT trial (2M-FS). The validation set for both models was ground truth labels from the first two minutes of data (i.e., the same set to enable direct FS comparison between models).

### 2.5. FS Model Evaluation

Five-fold cross validation was used to evaluate performance of both FS models. A temporal tolerance of ±2 frames (±0.04 s) was used to match ground truth manually labeled FS with predicted class labels. Evaluated metrics were sensitivity, specificity, accuracy, and precision.

Stride parameters were calculated using both manually labeled and automated FS. The difference between step time, stride time, and cadence from each group was calculated. These differences were compared to the minimal detectable change (MDC) for each stride parameter. Since MDC was not available for lower limb amputee gait, stride parameter MDC for healthy older adults was used [22,23,24].

### 2.6. Post-Processing

In [19], periods of repeated FS predictions corresponding with a single AP acceleration peak were observed in the preliminary data. This was also observed for the model trained on 2MWT data. Predicted FS labels were post-processed in MATLAB 2020b to correct for model prediction errors, including extra FS predictions and missed steps. Extra predictions were removed by identifying instances where two or more consecutive FS classifications occurred. The start and end of periods of consecutive predictions were located and the peak AP acceleration within the band was identified. The FS event corresponding to the AP peak was selected and all other predictions in this period were removed. To identify missed steps, periods where the duration between two consecutive steps was greater than 1.5 times the previous step were identified. An adaptive locking period was applied and searched for potential missed steps. Within the adaptive locking period, the AP acceleration peak was identified, and a FS was inserted at this timestamp. Final cleaned predictions were used for feature calculations.

### 2.7. Feature Calculations and Fall Risk Classification

A random forest machine learning model developed by Daines et al. [25] was used for amputee fall risk classification. Smartphone acceleration and angular velocity signals were used to calculate step-based feature sets, one using automated FS from the 6M-FS model, one using automated FS from the 2M-FS model, and a comparator set using manually labeled FS. In total, 62 features were extracted for each feature set (Table 2). Once features were extracted for each step, the minimum, maximum, mean, and standard deviation were calculated over all included steps for a total of 248 features (62 features multiplied by 4 statistics) per data set.

## 3. Results

A total of 12,308 foot strikes were identified and labeled in the first two minutes of walk test data, accounting for 3.06% of total output labels (402,000). Table 3 displays foot strike classification confusion matrices for the 6M-FS and 2M-FS models. The 6M-FS accuracy was 99.2%, sensitivity was 91.7%, specificity was 99.4%, and precision was 82.7%. The 2M-FS accuracy was 98.0%, sensitivity was 65.0%, specificity was 99.1%, and precision was 68.6%. The 2M-FS model FS classification was poor, with 35% of steps missed. Therefore, further analysis of stride parameters and fall risk classification was not possible using foot strikes from the 2M-FS model.

Automated FS, identified using the 6M-FS model, and manually labeled FS from the first two minutes of data were used to calculate stride parameters. The average and standard deviation difference between manual and automated FS stride parameters were calculated and compared to MDC values for these outcomes (Table 4).

Fall risk analysis was completed with foot strikes during the first two minutes of data, identified using the 6M-FS model, and the Daines et al. fall risk model [25] (Table 5). In total, 61 of 80 participants were correctly classified (76.3% accuracy, 48.1% sensitivity, 90.6% specificity). Classification using features from manually labeled FS resulted in 63 out of 80 participants correctly classified (78.8% accuracy, 51.9% sensitivity, 92.5% specificity).

## 4. Discussion

This research had two outcomes, FS identification and fall risk classification. The automated LSTM FS identification approach from [19], when trained on smartphone signals from six minutes of data and applied to two minutes of data, identified FS and non-FS events with 99.2% accuracy. When an LSTM was trained on smartphone signals from only the first two minutes of data and applied to two minutes of data, FS identification was poor (35% of steps missed), even after offline post-processing for error correction. The LSTM trained on six minutes of data outperformed the LSTM trained on two minutes of data in all FS identification performance metrics.

Further investigation into the results of the LSTM trained on two minutes of data (i.e., 2M-FS) revealed that, for 23 participants (~34% of participants), fewer than 50% of the average number of FS were detected. Furthermore, 7 of these 23 participants had fewer than 10 total steps detected by the LSTM. There were no identifiable similarities between these participants, and this sub-group included both people who were fall risk and people who were not at risk of falling. In this case, stride parameter analysis and step-based feature calculation for fall risk classification for these participants would not be feasible since most features are based on stride analysis. Therefore, to complete stride parameter analysis and fall risk classification for all participants, only automated FS identified in the first two minutes of data by the 6M-FS model was used.

LSTM is a deep learning approach that is often trained on large datasets of sequential data. Deep learning approaches are more complex than decision trees and other machine learning approaches and can often perform with higher accuracy on large datasets. However, deep learning often requires a greater amount of labeled data to prevent overfitting when training [27]. This could explain why the LSTM trained on the full six minutes of walk test data had better FS identification than the LSTM trained only on the first two minutes of walking; two minutes of walking for each participant was not enough data for an LSTM to isolate FS events. 

Step time, stride time, and cadence were calculated using automated FS from the 6M-FS model and using manually labeled FS from the first two minutes of data. The difference in stride time between automated FS and manually labeled FS calculations from the first two minutes of data was within the MDC for healthy older adults, while step time and cadence were outside of the MDC for healthy older adults. MDC for healthy older adults was used as a comparator since these values do not exist for lower limb amputees. Average step time and stride time for automated FS were both within approximately 0.04 s of the manually labeled FS, suggesting step time and stride time calculated using automated FS are comparable to manually labeled FS for most participants.

Fall risk classification was performed using a random forest model trained on step-based features calculated from automated FS identified in the first two minutes of data by the 6M-FS model. While the random forest correctly classified 76.3% of all participants and over 90% of non-fall risk participants, only 13 out of 27 (48.1%) fall risk participants were correctly classified. This means that ~52% of people who had fallen in the past six months were mistakenly classified as not being at risk of falling. In a clinical setting, this could translate to patients not being referred for further testing and a delay in the implementation of fall intervention strategies. These results indicated that step-based features from a 2MWT cannot be recommended for fall risk classification in lower limb amputees without further model development. Further refinement of the fall risk model using data from a 6MWT could improve classification results to a clinically usable standard. 

It is important to consider that this study used the first two minutes of a 6MWT to approximate a 2MWT. When observed in a clinical setting, some people begin walking at a comfortable pace with a typical gait pattern, but more gait deviations and instability become apparent as they continue walking. People may also walk faster if they know they only have to walk for two minutes instead of six minutes. This may explain why fall risk classification was worse when features were calculated from two minutes of data than six minutes of data; the signals collected during the first two minutes of data were not sufficient to differentiate fall risk participants and non-fall risk participants. 

It is possible that data from the full six minutes may not be required to differentiate fallers from non-fallers. In previous fall risk classification research, random forest models trained on signals from 6MWT trials focused on signal data during turns only. For example, Drover et al. [28] classified fall risk in older adults from 6MWT data collected from multiple sensors with 77.3% accuracy, 66.1% sensitivity, and 84.7% specificity and noted that turn data improved all classification metrics. In addition, Daines et al. [25] used turn data from 6MWT trials in lower limb amputees with 81.3% accuracy, 57.2% sensitivity, and 94.9% specificity, however, manual FS and turn identification was required. Future research could examine if features calculated from automated FS during turns from 2MWT (and 6MWT) would improve fall risk classification. Additionally, differences in gait characteristics that distinguish fallers and non-fallers may become more apparent as the test progresses. Step-based features calculated from automated FS during the final minutes of the walk test could be analyzed as a possible improvement on the fall risk classification model. Future studies could also investigate other gait-based features that may better differentiate fallers from non-fallers in a shorter walk test. 

Fall risk information is not normally available from 2 or 6MWT, so this is promising for the future development of an AI-enhanced TOHRC Walk test app for use in lower limb amputees without requiring separate models integrated into the smartphone application for each walk test. Stride parameters can be automatically calculated from the automated FS for either a 6MWT or 2MWT immediately after completing the walk test with reasonable confidence. A model for fall risk classification could be integrated into a future application and applied to 6MWT data, however, it is not recommended for 2MWT data. 

## 5. Conclusions

Foot strike identification is essential to define the gait cycle and calculate stride parameters. AI tools for clinical analysis (e.g., fall risk classification) rely on proper gait segmentation to calculate step-based features. This research determined that a smartphone app can provide accelerometer and gyroscope signals during a 6MWT or 2MWT for AI-based analyses to automatically determine foot strikes. The FS identification model used a LSTM deep learning approach trained on six minutes of data, with this model being applicable for identifying FS in both six minutes of data and two minutes of data with at least 99% accuracy. However, a model only trained on the first two minutes of data had poor foot strike identification results, thereby not supporting use of this approach for outcome measurement. 

Step and stride time calculated using automated FS in the first two minutes of data identified by the 6M-FS model of smartphone data were equivalent to manually labeled FS for most participants, indicating that the 2MWT stride outcomes measurements could be viable for clinical analysis. Integration of this FS detection model into the TOHRC Walk Test app could allow for immediate stride parameter analysis in lower limb amputees after completing a 6MWT or 2MWT. However, fall risk classification using step-based features calculated from automated FS is not recommended for the 2MWT.

## Figures and Tables

**Figure 1 sensors-22-01749-f001:**
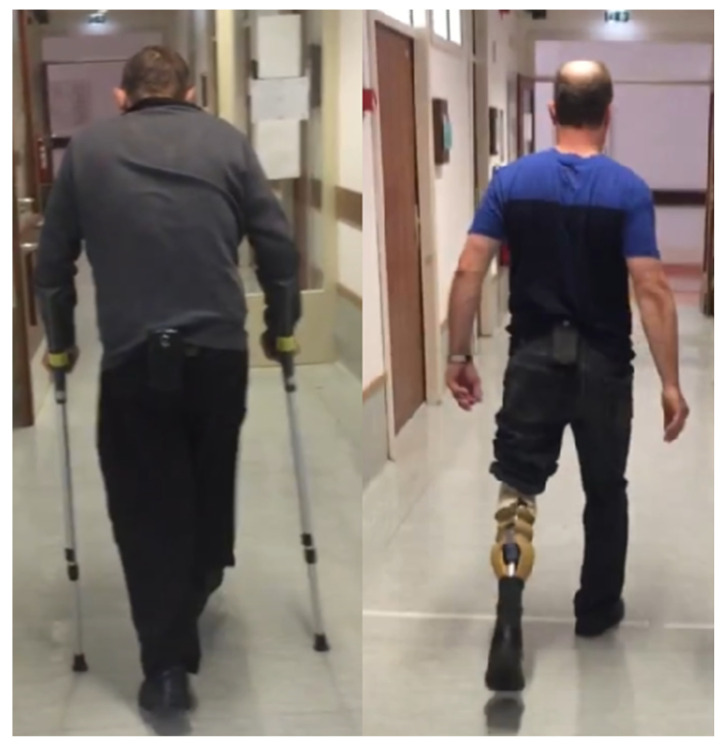
Experimental set-up: smartphone on posterior pelvis.

**Figure 2 sensors-22-01749-f002:**
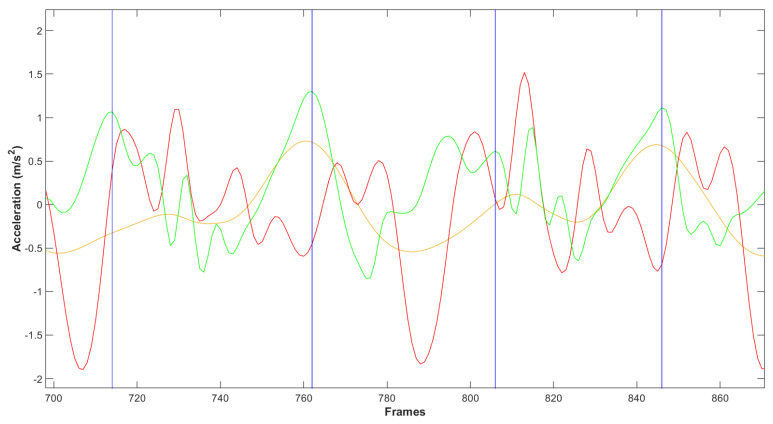
Filtered smartphone signals over time. Medio-lateral acceleration (yellow curve), vertical acceleration (red curve), and anterior–posterior (AP) acceleration (green curve) were used to identify ground truth foot strikes. Typically, foot strikes correspond with an AP acceleration peak followed by a vertical acceleration peak. Video recording of the trial was used to confirm the timestamp of foot strikes. Vertical blue lines indicate frames manually identified as ground truth labels.

**Table 1 sensors-22-01749-t001:** Participant characteristics.

Characteristic	Value
Age (years)	64.2 ± 12.2 (19–90)
Sex	
Male	63 (78.8%)
Female	17 (21.2%)
Fall risk status	
Fall risk	27 (33.8%)
No fall risk	53 (66.2%)
Level of amputation	
Transtibial	72 (90.0%)
Transfemoral	3 (3.8%)
Bilateral (transtibial)	5 (6.2%)
Time since amputation (years)	15.7 ± 18.0 (0–65)
Ambulatory aid use	
No aids	42 (52.5%)
Double crutches	25 (31.3%)
Single cane/crutch	12 (15.0%)

Note: Data are presented as mean ± SD (range) or number (percentage).

**Table 2 sensors-22-01749-t002:** Feature list.

Temporal	Descriptive Statistics	Frequency Domain Features
Cadence	Minimum ML	Quartile FFT ML
Step time right	Minimum AP	Quartile FFT AP
Step time left	Minimum Vert	Quartile FFT Vert
Stride time	Maximum ML	Quartile FFT Tilt
Symmetry index	Maximum AP	Quartile FFT Rotation
	Maximum Vert	Quartile FFT Obliquity
	Mean ML	Maximum FFT ML
	Mean AP	Maximum FFT AP
	Mean Vert	Maximum FFT Vert
	Mean Tilt	Maximum FFT Tilt
	Mean Rotation	Maximum FFT Rotation
	Mean Obliquity	Maximum FFT Obliquity
	Range Tilt	Standard Deviation FFT ML
	Range Rotation	Standard Deviation FFT AP
	Range Obliquity	Standard Deviation FFT Vert
	Standard Deviation ML	Standard Deviation FFT Tilt
	Standard Deviation AP	Standard Deviation FFT Rotation
	Standard Deviation Vert	Standard Deviation FFT Obliquity
	Standard Deviation Tilt	Peak Distinction FFT ML
	Standard Deviation Rotation	Peak Distinction FFT AP
	Standard Deviation Obliquity	Peak Distinction FFT Vert
	RMS ML	Peak Distinction FFT Tilt
	RMS AP	Peak Distinction FFT Rotation
	RMS Vert	Peak Distinction FFT Obliquity
	RMS Tilt	REOH ML
	RMS Rotation	REOH AP
	RMS Obliquity	REOH Vert
		REOH Tilt
		REOH Rotation
		REOH Obliquity

Symmetry index: symmetry in right and left limb step times [26]. AP: anterior–posterior; ML: medio-lateral; RMS: root-mean square; FFT: fast Fourier transform; REOH: ratio of even/odd harmonic frequencies.

**Table 3 sensors-22-01749-t003:** Foot strike classification.

6M-FS			2M-FS		
	Foot Strike	No Foot Strike		Foot Strike	No Foot Strike
Foot strike	11,283	1025	Foot strike	8006	4302
No foot strike	2361	386,010	No foot strike	3672	383,200

**Table 4 sensors-22-01749-t004:** Average and standard deviation (in brackets) difference between manual and automated foot strike stride parameter outcome measures for the 6M-FS model. MDC = minimum detectable change.

	Automated FS	MDC
Step time (s)	0.045 (0.11)	0.042
Stride time (s)	0.044 (0.09)	0.772
Cadence (steps/min)	−28.91 (37.19)	8.44

**Table 5 sensors-22-01749-t005:** Fall risk classification confusion matrices for automated and manual foot strike (FS) identification.

Automated FS			Manual FS	
	Fall Risk	No Fall Risk	Fall Risk	No Fall Risk
Fall risk	13	14	14	13
No fall risk	5	48	4	49

## Data Availability

Not applicable.

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
