# Peer review of "Amputee Fall Risk Classification Using Machine Learning and Smartphone Sensor Data from 2-Minute and 6-Minute Walk Tests"

_sensors, 2022, doi:10.3390/s22051749_

Round 1

Reviewer 1 Report

This Reviewer appreciates the opportunity for reviewing this paper. In this study, the authors investigated automated foot strike (FS) detection and fall risk classification using signals from 2-minute walk test (2MWT) and 6-minute walk test (6MWT). A long short-term memory (LSTM) model was used for automated foot strike detection. A random forest machine learning model was used for amputee fall risk classification. Features calculated using automated FS from a 2MWT could not sufficiently classify fall risk in lower limb amputees, and are only recommended for the 6MWT. Overall, the manuscript is well organized and well-written. The Reviewer has a couple of comments as follows:

  1. Abstract: The LSTM was used for the FS detection, and a random forest machine learning model was used for fall risk classification. However, only LSTM was mentioned in the Abstract. The Reviewer suggests adding some descriptions of the random forest machine learning model in the Abstract.
  2. Abstract: The abbreviation “FS” was mentioned at the first time without full name, which may cause confusion. The Reviewer suggests adding the full name.
  3. Methods (P4): “… a simulated 2MWT dataset was created. The first two minutes of the 6MWT trial was determined to most closely replicate that of a 2MWT, so the first two minutes of each participant’s trial was exported for a total of 6000 data points per participant”. I wonder if another simulated 2MWT dataset, containing the first one minute and the last one minute of the 6MWT, is more closely replicate that of a 2MWT, since it contains a start process and an end process. Is it possible to use this simulated 2MWT dataset to train the model and see the results? If it takes too long time to re-do the training process, the Reviewer is okay with only adding some discussions in this study based on expectations and hope it could be included in a future study.
  4. Methods (P5): “The LSTM model was imported from Keras”. The Reviewer suggests describing the fundamental algorithm of the LSTM model, probably with a flow chart. Moreover, a reference (e.g., the website link) may be added for the Keras.
  5. Methods (P5): it seems that there is an extra space after Refs. [22-24].

Reviewer 2 Report

  1. When reading the abstract, it is difficult to establish what the main paper contribution is. The authors mention the development of a LSTM model and the proposal of a novel methodology for automated foot strike identification but, they are not reported with detail throughout this paper. 
  2. In section 2.2, is sampling rate 50Hz?

    Section 2.3.1 contradicts this. Here it is affirmed that sampling rate of the smartphone is not a constant so, an interpolation was required, and the resulting signal was sampled at 50Hz.

    Provide references (not attributable to this paper authors) or any analysis that shows that there is no loss of information in the signal or significative distortion due to variable sampling rate.

    Please provide details of the interpolation method used and how the 50Hz resampled signal matches the original signal. Explain in detail how you reconstruct the signal (use of interpolation filter, zero-order  holder, etc).

    Provide any reference (not attributable to this paper authors) or analysis that shows that signal components lower than 4Hz are useful to determine foot strikes.

  3. Provide figures that show the signals used to determine FS 

  4. Sections 2.2 and 2.3.2 are completed contained in section 2.2 and 2.4.2 of reference 19. 
  5. Respect to reference 20, please do not use  not published references.
  6. Please provide a comparative table that shows the result of this work calculating the fall risk and other works reported in the literature. 

Round 2

Reviewer 1 Report

Authors have satisfactorily addressed my comments

Author Response

No further revisions for this reviewer.

Reviewer 2 Report

Please, make sure that when you submit your response, you remove the revision comments from the authors.

1.  You should specify if you re-constructed your signal first or filtered the signal first. It is not clear in section 2.3.1. By the other hand, in the authors’ response document that you submitted, you state that you filter the signal first and then you interpolate the signal. This is a great concern for me because:

 Is your filter designed to process nonuniform sampling rate signal? Or is it a uniform sampling rate signal filter?

  1. I do not agree with NB1’s review comment, I was not confused since an important part of your work is the use of “smartphone sensors”. You should show that you are considering the adequate methodology to overcome the sampling problems that these devices have. I don’t agree with the last two lines of NB1’s comment, I’m very sure you implemented this before and that it worked. Any interpolation process might distort your signal and you could have important information losses and sometimes your device could fail with no explanation.

Anyone else reading your work might have the same questions about your filtering and processing metodology, please provide details (so the reader doesn’t assume) about your filtering and interpolation methodology, and it would be great if you could provide a figure showing a segment of your original signal, interpolated signal, and filtered signal so it could be veryfied that no major distorsión is present.
